



# Using the Galilean Relativity Principle to Understand the
# Physical Basis for Magnetosphere-Ionosphere Coupling
# Processes
Anthony J. Mannucci[1], Ryan McGranaghan[2], Xing Meng[1], Bruce T. Tsurutani[1] and Olga P. Verkhoglyadova[1]
[1]Jet Propulsion Laboratory, California Institute of Technology, Pasadena, CA, USA
[2]Atmospheric and Space Technology Research Associates (ASTRA), Boulder CO, USA
*Correspondence to:* Anthony J. Mannucci (anthony.j.mannucci@jpl.nasa.gov)
**Abstract** We use the Principle of Galilean Relativity (PGR) to gain insight into the physical basis for
magnetosphere-ionosphere coupling. The PGR states that the laws of physics are the same in all inertial
reference frames, considering relative speeds between such reference frames that are significantly less than
the speed of light. The PGR is a limiting case of the principle of Special Relativity, the latter applicable to
any relative speeds between two inertial reference frames. Although the PGR has been invoked in past works
related to magnetosphere-ionosphere coupling, it has not been fully exploited for the insights it can provide
into such topics as large-scale ionospheric convection and high latitude heating. In addition, the difficulties
of applying the PGR to electrodynamics has not been covered. The PGR can be used to show that in the high
latitude ionosphere there often exists a reference frame where electric fields vanish at lower altitudes where
collisions are important (altitudes near ~100-120 km). In this reference frame, it is problematic to assert that
currents of magnetospheric origin cause horizontal electric fields in the ionosphere, as has been suggested
for the causal origin of Subauroral Polarization Stream electric fields. Electric fields have also been invoked
as the causal origin of large-scale ionospheric convection, which may be a problematic assertion in certain
reference frames. The PGR reinforces the importance of the neutral species and ion-neutral collisions in
magnetosphere-ionosphere coupling, which has been noted by several authors using detailed multi-species
plasma calculations. A straightforward estimate shows that the momentum carried by electron field aligned
currents of magnetospheric origin during disturbed periods is much less than the momentum changes
experienced by the neutral species in an Earth-fixed frame. The primary driver of neutral species momentum
changes during disturbed periods is the momentum imparted by the solar wind to ionospheric ions resulting
from electrodynamic interactions. This is consistent with the idea that electric fields do not lead to large scale
ionospheric convection.





## 1 Introduction

Electrodynamics as it pertains to magnetosphere-ionosphere coupling is a critical aspect of the ionospheric
response during geomagnetic storms.  Large scale convection of the high-latitude ionospheric plasma (auroral
latitudes and higher), and heating of the plasma and neutral species during disturbed conditions is a
consequence of electric and magnetic forces that change dramatically when solar wind conditions lead to
geomagnetic storms. Scientific consensus on fundamental aspects of the physical processes that occur at high
latitude is not yet achieved, including the definition of Joule heating (Vasyliunas and Song, 2005;
Verkhoglyadova et al. 2017). In this paper, we are able to gain insight into these physical processes by using
the simple but powerful Principle of Galilean Relativity (PGR), which states that physical laws are invariant
with respect to inertial reference frame.

Despite its deceptively simple expression, the PGR has important implications for high latitude
electrodynamics. This arises due to the inertial reference frame-dependent property of the electric field in the
high-latitude ionosphere. The large-scale electric field is directed predominantly in the horizontal direction
and the magnetic field tends to be predominantly in the vertical direction. The PGR requires that electric
fields in the high latitude ionosphere vary substantially according to inertial reference frame. This fact has
been appreciated in the literature but not fully exploited for its physical implications. In particular, the concept
that field-aligned currents cause electric fields that lead to high velocity plasma flow has been invoked for
high latitude phenomena (Anderson et al., 1993). However, if another inertial reference frame is chosen,
these same arguments would seem problematic because the electric field may vanish. The notion of a
"preferred inertial reference frame" for high latitude electrodynamics is often cited (Vasyliūnas and Song,
2005; Leake et al., 2014; Strangeway, 2012) and is useful when considering that high latitude phenomena
occur within the physical media of plasma and neutral gases. However, a reference frame where the electric
field vanishes is also useful to consider in understanding the physical basis of magnetosphere-ionosphere
coupling.

In this paper, we discuss how the PGR affects high latitude electrodynamics. We discuss the literature on the
topic of Galilean electrodynamics, which is the low-velocity limit of the theory of special relativity applied
to electrodynamics. Galilean electrodynamics is used in the literature of magnetosphere-ionosphere coupling,
but incompletely in the sense that transformation of the source terms – charges and currents – are not
considered along with the field transformation equations. This leads to a contradictory set of equations
whereby magnetic fields do not change between inertial reference frames, but currents do, even though
currents are the source term for magnetic fields. We review how the PGR has been referred to in the literature
and how it can be used to help interpret physical processes. We next discuss how the frame-variant nature of





the electric field can be used to interpret the physical basis of Ohm's law and high latitude electromagnetic
energy conversion. Finally, in seeking physical explanations that are robust to choice of inertial reference
frame, we are led to de-emphasize electric fields as a source of ion motion and electron currents at high
latitudes. We discuss how the literature emphasizes the importance of relative velocities between different
species populations within the plasma as the root cause of high latitude changes during disturbed periods.
The question presents itself: what causes the ions to convect and acquire a different velocity than the neutral
species? We address this question.

## 2 The Principle of Galilean Relativity in Electrodynamics


The Principle of Galilean Relativity (PGR) determines how physical quantities change between inertial
reference frames traveling at speeds significantly less than the speed of light. These quantities include electric
and magnetic fields, and their sources such as charges and currents. The PGR is a limiting case of the principle
of Special Relativity, the latter being applicable for any relative speeds between inertial reference frames.
Either form of relativity is an important symmetry of nature: physical laws do not depend on one's velocity
relative to an "absolute" or preferred reference frame.

What makes the PGR a useful idea in the context of high latitude electrodynamics is that the electric field
component perpendicular to $\mathbf{B} - \mathbf{E}$ being largely horizontal at high latitudes – substantially changes with
choice of inertial reference frame, even for relatively low velocities that are characteristic of high latitude
processes (e.g. ~100s of m/s to a few km/s). The PGR can help to interpret physically the equations governing
high latitude electrodynamics by considering these equations in different inertial reference frames.

In this section, we review the literature of how electric and magnetic fields transform in the low velocity limit
that corresponds to the PGR. Throughout the text, we use the symbol $\mathbf{v_r}$ to indicate the relative velocity
between two inertial reference frames, for example frames $\mathcal{A}$ and $\mathcal{B}$. We use primed variables to refer to
physical quantities in the inertial reference frame $\mathcal{B}$ moving with velocity $\mathbf{v_r}$ relative to reference frame $\mathcal{A}$.

As first deduced by Einstein in 1905, electric and magnetic fields transform between inertial reference frames
according to the following relationships (Rousseaux, 2014; Heras, 2010):

$$\mathbf{E}' = \gamma \left( \mathbf{E} - \frac{\gamma - 1}{\gamma} \frac{\mathbf{v_r}(\mathbf{v_r} \cdot \mathbf{E})}{v_r^2} + \mathbf{v_r} \times \mathbf{B} \right) \qquad (1)$$






$$\mathbf{B}' = \gamma\left(\mathbf{B} - \frac{\gamma - 1}{\gamma}\frac{\mathbf{v}_r(\mathbf{v}_r \cdot \mathbf{B})}{v_r^2} - \frac{1}{c^2}\mathbf{v}_r \times \mathbf{E}\right) \tag{2}$$


where $(\mathbf{E}', \mathbf{B}')$ are the electric and magnetic fields in the new inertial reference frame, $(\mathbf{E}, \mathbf{B})$ are the fields in
the original frame, and $\gamma = 1/\sqrt{1 - v_r^2/c^2}$ with $c$ being the speed of light.

It is clear from these equations that there is not a unique low-velocity limit applicable to $(\mathbf{E}, \mathbf{B})$ because the
field transformations do not depend on velocity exclusively, but also on the electric and magnetic fields
themselves. Le Bellac and Lévy-Leblond (1973) discuss two low-velocity limiting cases, which have since
spawned a literature on the topic of "Galilean electromagnetism". These are the electric and magnetic limits,
according to which field magnitude is dominant. The electric limit applies when $E \gg cB$, and the magnetic
limit applies when $cB \gg E$. High latitude electrodynamics encompasses the magnetic limit, which
corresponds to the fact that Earth's magnetic field is relatively strong and plasmas are quasi-neutral: the
sources of electric fields (charges) are generally neglected compared to the sources of magnetic fields
(currents). Magnetic fields in a plasma are caused by currents arising in a quasi-neutral medium because
positive and negative charges move in opposing directions.

In the magnetic limit ($cB \gg E$) and assuming that terms of order $v_r/c$ are small, the Lorentz transformation
laws (Equations (1) and (2)) become (Le Bellac and Lévy-Leblond, 1973):

$$\mathbf{E}' = \mathbf{E} + \mathbf{v}_r \times \mathbf{B} \tag{3}$$


$$\mathbf{B}' = \mathbf{B} \tag{4}$$


which are familiar transformation rules in the context of space physics (e.g. Parks, 2007; Vasyliūnas and
Song, 2005). We refer to these low-velocity limit equations as comprising a Galilean transformation, by
analogy to the more general Lorentz transformation.

An alternative derivation of the Galilean transformation of electric and magnetic fields is possible by
considering the Lorentz force law (Preti et al., 2009; Heras, 2010). The Lorentz force $\mathbf{F}$ in inertial reference
frame $\mathcal{A}$ is given by:

$$\mathbf{F} = q(\mathbf{E} + \mathbf{v} \times \mathbf{B}) \tag{5}$$


where $\mathbf{F}$ is the force on a charge $q$ moving with velocity $\mathbf{v}$ in frame $\mathcal{A}$, and where the electric field is $\mathbf{E}$ and
the magnetic flux density is $\mathbf{B}$. If applied to a charged particle of mass $m$, the Lorentz force will result in
acceleration $\mathbf{F}/m$. This acceleration is independent of inertial reference frame. Therefore, if $\mathbf{F}'$ represents the



force measured in an inertial reference frame $\mathcal{B}$ moving with velocity $\mathbf{v}_r$ with respect to the original frame
$\mathcal{A}$, we know that $\mathbf{F}' = \mathbf{F}$. In reference frame $\mathcal{B}$ the particle velocity is $\mathbf{v}' = \mathbf{v} - \mathbf{v}_r$, and the particle's mass
and charge are invariant with respect to inertial reference frame (Galilean approximation). The equality of
forces between the two inertial reference frames requires that:

$$(\mathbf{E} + \mathbf{v} \times \mathbf{B}) = (\mathbf{E}' + (\mathbf{v} - \mathbf{v}_r) \times \mathbf{B}') \tag{6}$$


which is achieved if Equations (3) and (4) are used.

Equation 3 shows that only the component of the electric field parallel to $\mathbf{B}$ is unchanged under a Galilean
transformation (i.e. $\mathbf{B}$ is Galilean invariant, GI), whereas the perpendicular electric field changes depending
on the relative velocity of frame $\mathcal{B}$. The frame-variant nature of the electric field has significant implications
within the context of high-latitude electrodynamics, as we discuss below. Table 1 summarizes how different
physical quantities relevant to high latitude electrodynamics vary under a Galilean transformation in the
magnetic limit.

The source terms of the fields, charge density $\rho$ and current density $\mathbf{J}$, also transform according to the principle
of special relativity. As shown by Rousseaux (2014), they transform as a four vector according to:

$$\mathbf{J}' = \mathbf{J} - \gamma \mathbf{v}_r \rho + (\gamma - 1) \frac{\mathbf{v}_r (\mathbf{v}_r \cdot \mathbf{J})}{v_r^2} \tag{7}$$


$$\rho' = \gamma \left( \rho - \frac{(\mathbf{v}_r \cdot \mathbf{J})}{c^2} \right) \tag{8}$$



In the magnetic limit, the transformation equations become (Le Bellac and Lévy-Leblond, 1973):

$$\rho' = \rho - \frac{(\mathbf{v}_r \cdot \mathbf{J})}{c^2} \tag{9}$$


$$\mathbf{J}' = \mathbf{J} \tag{10}$$


The magnetic limit is associated with the condition $c\rho \ll J$. The Galilean invariance of current expressed by
Equation (10) is familiar in the context of ionospheric electrodynamics (e.g. Thayer and Semeter, 2004) and
is intuitive when charge density is zero. (Currents in the presence of no charge density arise from oppositely
charged particles moving in opposite directions). The invariance of currents is consistent with the Galilean





invariance of magnetic fields. We note that zero charge density in the original reference frame leads to a
small charge density in the moving frame according to Equation (9). The literature of ionospheric
electrodynamics admits of non-zero charge densities or "charge accumulation" (Figure 3 in Vasyliunas,
2012) leading to "polarization electric fields" (Richmond and Thayer, 2000) that is inconsistent with the
transformation law Equation (4). In a moving frame the net charge becomes a current that must lead to
modification of the magnetic field. Thus, electric fields that arise due to charge accumulation are inconsistent
with the usual field transformation equations adopted in the space physics literature.
In Figure 1 we depict an idealized situation where electric and magnetic fields are at perpendicular angles to
each other, to represent approximately the high latitude ionosphere where the Earth's magnetic field is close
to vertical and large-scale convection electric fields are predominantly horizontal. We will refer to this
geometry elsewhere in the text. For electric fields of magnitude ~50 mV/m, which can occur at high latitude
in a reference frame rotating with the Earth, and high latitude magnetic field magnitudes of ~50,000 nT (near
120 km altitude), the electric field will be nearly zero in a reference frame moving with a speed of ~1.6 km/s
relative to the Earth. The necessary direction of such a moving frame is shown in Figure 1. For situations
relevant to high latitude electrodynamics, an electric field that is non-negligible observed from an Earth fixed
frame can be zero viewed from an inertial reference frame moving at speeds consistent with the PGR and the
transformation rules of Table 1. Although an Earth fixed frame is not inertial, the small acceleration in this
frame is typically ignored for the purposes of these estimates.
Table 1: Transformation of physical quantities under change of inertial reference frame in the Galilean
magnetic limit. The primed quantities are in the new reference frame moving at velocity $\mathbf{v}_r$ relative to the
original frame. All quantities are assumed to be quasi-static or slowly varying. Galilean invariance refers to
quantities that are the same in all inertial reference frames.

| Physical Quantity | Transformation | Comment |
|---|---|---|
| Electric field **E** | $\mathbf{E}' = \mathbf{E} + \mathbf{v}_r \times \mathbf{B}$ | The electric field in a direction parallel to **B** is invariant, as is the electric field in the absence of a magnetic field |
| Magnetic field **B** | $\mathbf{B}' = \mathbf{B}$ | |
| Current density **J** | $\mathbf{J}' = \mathbf{J}$ | Assumes no net charge. Not invariant if charges are present. |
| Charge density $\rho$ | $\rho' = \rho - \dfrac{1}{c^2}\mathbf{v}_r \cdot \mathbf{J}$ | Charge per unit volume. |
| Heat energy **Q** | $\mathbf{Q}' = \mathbf{Q}$ | Heat energy and temperature are Galilean invariant (GI) |
| Velocity **v** | $\mathbf{v}' = \mathbf{v} - \mathbf{v}_r$ | |



| Relative velocity between two species | Invariant | |
|---|---|---|
| Collisional forces | Invariant | Depends on relative velocities, which are invariants |



There are several implications of the electric and magnetic field transformation rules in Table 1. First,
Maxwell's equations are not invariant under these transformations, unless the displacement current term is
neglected in Ampere's law (Le Bellac and Lévy-Leblond, 1973; Preti et al., 2009; Heras, 2010). Second,
currents in a moving reference frame that arise from accumulated charges in the original frame do not
generate magnetic fields (Le Bellac and Lévy-Leblond, 1973). Fortunately, the quasi-neutral plasmas
characteristic of geospace do not lead to large errors because of the small magnitudes of such charges.

The PGR is mentioned in textbooks on electrodynamics such as Jackson (1975) and Pollock and Stump
(2001), primarily to show how the PGR fails in the context of electromagnetism. A point is made, however,
that the physical principle of relativity demonstrates that "**E** and **B** have no independent existence" (Jackson,
1975), which is true for Galilean as well as Special relativity.

Galilean relativity has been invoked within plasma physics in the context of wave-particle interaction and
Landau damping (Chen, 2016; Dawson, 1961). Electrons gaining or losing energy from a plasma wave
depends on the relative velocity of the electrons and the wave's phase velocity, hence the possibility of the
plasma waves to alter the velocity distribution function of the electrons. Although relative velocity is a
Galilean invariant, and hence so is the damping, the usual mathematical description of Landau damping does
not explicitly contain relative velocities, so it appears superficially that Landau damping might depend on
reference frame. (This is similar to how it superficially appears that the Lorentz force depends on reference
frame). The authors resolve this apparent paradox by showing how a reference frame change affects both the
electron distribution function and the wave energy, thus preventing Landau damping from violating the PGR.

**2.1 The PGR in Geospace**
Discussions of the PGR in the context of geospace are highly diverse. On the one hand, widely-used textbooks
that describe ionospheric electrodynamics (Kivelson and Russell, 1995; Kelley, 2009; Brekke, 2013) do not
refer to the PGR. In the journal-based literature, several authors discuss inertial reference frames in the
context of high latitude electrodynamics (examples are provided in the text below). The reference frame-
dependent property of the electric field is mentioned on occasion, but not emphasized or exploited in many
cases. To our knowledge, inconsistency between the full set of Maxwell's equations and the Galilean



transformation laws for electrodynamics has not been emphasized in the context of geospace. The
transformation of source terms is generally not discussed.

It should be noted that a "preferred reference frame" is a useful construct in plasma physics because of the
importance of material media that obey the laws of classical mechanics, such as the plasma and the neutral
atmosphere (Vasyliunas and Song, 2005; Leake et al., 2014). Song et al. (2001) derive different versions of
Ohm's law appropriate to different reference frames. The expressions for the conductivities depend on
whether one is in the inertial reference frame of the plasma or of the neutral species. This is related to the
discussion in Jackson (1975, Section 11.1) on why the wave equation for sound waves depends on velocity
relative to the medium carrying the waves. For electromagnetic waves, of course, there is no such medium
and no preferred reference frame.

An important construct to examine from the perspective of the PGR, and widely seen in the literature, is
based on the following quantity: $\mathbf{E} + \mathbf{V} \times \mathbf{B}$, where $\mathbf{E}$ is an electric field, $\mathbf{V}$ is the velocity of a constituent of
the material medium (ions, electrons, neutrals, etc.), and $\mathbf{B}$ is the magnetic field. In several publications the
quantity $\mathbf{E} + \mathbf{V} \times \mathbf{B}$ is referred to as "the electric field in the reference frame of species X" where $\mathbf{V}$ is the
bulk velocity of that species (Vasyliūnas, 2012; Vasyliūnas and Song, 2005; Thayer and Semeter, 2004;
Richmond, 1995; Leake et al., 2014; Hidekatsu, 2009; Strangeway, 2012). The symbol $\mathbf{E}'$ is often used to
denote this quantity, but in this work, we will use the convention $\mathbf{E}^* = \mathbf{E} + \mathbf{V} \times \mathbf{B}$ (following Vasyliūnas
and Song, 2005) to maintain use of the prime symbol to refer to transformations between inertial reference
frames.

Using the PGR, it is immediately clear that $\mathbf{E}^*$ is not an electric field because of its transformation properties.
$\mathbf{E}^*$ is a Galilean invariant, whereas an electric field depends on inertial reference frame. For similar reasons,
the term $\mathbf{V}_n \times \mathbf{B}$ is not an electric field, although it has been referred to as a dynamo electric field, where $\mathbf{V}_n$
in the velocity of the neutral species (Richmond, 1995).

The GI property of $\mathbf{E}^*$ holds no matter what the velocity $\mathbf{V}$ refers to, whether that of the neutral species, or
the ions, etc. The use of $\mathbf{E}^*$ has encouraged the exploration of reference frames tied to material media (e.g.
Leake et al., 2014), whereas for electrodynamic quantities there is no need for a preferred inertial reference
frame. In many situations applicable to the high latitude ionosphere, *there exists a reference frame for which*
*the electric field is zero* (Figure 1). This frame is not tied to any particular medium, but is instructive to
consider. As we show in the discussion of Ohm's law, currents can arise in the absence of electric fields for
precisely the reason that currents depend on $\mathbf{E}^*$ rather than $\mathbf{E}$. Considering this special inertial reference frame
– where the electric field vanishes – provides insight into the physical basis for momentum and energy
changes at high latitudes.




## 2.2 The PGR and Ohm's Law

Ohm's law is derived from the plasma force balance equations taking collisions into account (Song et al.,
2001; Richmond, 1995). These equations can be written, for ions and electrons, as:

$$qN_e(\mathbf{E} + \mathbf{u}_i \times \mathbf{B}) = N_e m_i v_{in}(\mathbf{u}_i - \mathbf{u}_n) + N_e m_i v_{ie}(\mathbf{u}_i - \mathbf{u}_e) \qquad (11)$$


$$-qN_e(\mathbf{E} + \mathbf{u}_e \times \mathbf{B}) = N_e m_e v_{en}(\mathbf{u}_e - \mathbf{u}_n) - N_e m_e v_{ei}(\mathbf{u}_i - \mathbf{u}_e) \qquad (12)$$


Charge neutrality is assumed such that $N_e$ is the charge density of either electrons or ions. $q$ is the elementary
charge, $\mathbf{u}_e$, $\mathbf{u}_n$ and $\mathbf{u}_i$ are the electron, neutral and ion velocities, respectively, and $m_i$ and $m_e$ are the ion and
electron masses, respectively. We assume a single ion species for simplicity. $v_{in}, v_{ie}, v_{en}$, and $v_{ei}$ are the ion-
neutral, ion-electron, electron-neutral and electron-ion momentum transfer rates, respectively. These rates
are sometimes referred to as "collision rates". The following reciprocity relation applies to these rates:
$m_k v_{kl} = m_l v_{lk}$, where $k, l$ represent one of the species: electron, ion, or neutral (see Gombosi, 2004), and
$k \neq l$. We are ignoring pressure forces and other forces such as gravity, centrifugal, etc. Since these are force
balance equations, the acceleration of each species is zero. It is clear these equations are consistent with PGR
and are valid in any inertial reference frame, as can be readily seen by applying the transformations from
Table 1. A simplification that collision frequencies do not depend on relative velocities is assumed; see
Richmond (1995) for a statement regarding this limitation.

A form of Ohm's law that is derived from the force balance equations is as follows (see derivation provided
by Song et al., 2001):

$$\mathbf{J} = \sigma_\parallel \mathbf{E}_\parallel + \sigma_P(\mathbf{E}_\perp + \mathbf{V} \times \mathbf{B}) + \sigma_H \mathbf{B} \times (\mathbf{E} + \mathbf{V} \times \mathbf{B})/B \qquad (13)$$


where the current density $\mathbf{J}$ is given by $qN_e(\mathbf{u}_i - \mathbf{u}_e)$ (charge neutrality is assumed). $\sigma_\parallel, \sigma_P$ and $\sigma_H$ are the
parallel, Pedersen and Hall conductivities, respectively. $\mathbf{V}$ might be the bulk plasma velocity or the neutral
wind velocity (see Song et al., 2001, Equation (17). In our Equation (13), we have corrected typographical
errors of Song et al.'s (2001) Equation 17). The conductivities typically involve terms that are GI, such as
collision frequencies and gyrofrequencies that depend on the invariant magnetic field. Since conductivities
are GI, this form of Ohm's law is explicitly GI, since both the left-hand side and right-hand side do not
change under the Galilean transformation. In fact, any velocity $\mathbf{V}$ ensures that the right hand side of Eq. (13)
is GI.

Ohm's law Equation (13) states that that currents can be generated by 1) electric fields alone, 2) neutral winds
alone, or 3) a combination of the two. The choice of inertial reference frame influences these three different
possibilities. The causal connection between electric fields and currents is reference frame dependent in the



collisional ionospheric plasma where Ohm's law applies. If an inertial reference frame is chosen such that
the perpendicular electric field is zero, then in that reference frame the currents are not generated by electric
fields. Strangeway (2012) also questions whether electric fields cause the currents, instead suggesting that
both the electric field and the currents are a consequence of the plasma flow. From the perspective of the
PGR, it must be concluded that it is inconsistent to assert that electric fields cause plasma flow, at least in
certain reference frames. It is consistent with the PGR to suggest that flow differences between plasma and
neutrals are responsible for currents.
Further clarification is found in Vasyliunas (2012) work on the "physical basis for ionospheric
electrodynamics". Vasyliunas (2012) suggests that the ionospheric current is primarily a stress-balance
current ultimately due to the relative motion between plasma and neutrals. That paper thus makes the direct
link between two GI quantities, currents and relative velocities, without the problematic intermediary of the
reference frame-dependent electric field. In the discussion section, we remark on the implications of requiring
a direct relationship between currents and electric fields, in the context of Subauroral Polarization Stream
(SAPS) electric fields.
The PGR is also relevant to Figures (3) and (4) in Vasyliunas (2012) which diagram two differing physical
interpretations of the neutral wind dynamo. We note that the interpretation on the left, the "conventional
approach", requires the creation of polarization charges, which is inconsistent with the Galilean invariance
of the magnetic field (Equation (4)). The charges generate currents in a moving reference frame, thus
modifying the magnetic field in the moving frame.
**2.3 The PGR and Poynting's Theorem**
Poynting's theorem (PT) and Poynting flux are often used in the context of high latitude electrodynamics to
understand energy deposition (Kelley, 1989; Thayer and Semeter, 2004). In this section, we discuss how
Poynting's theorem is incompatible with Galilean electromagnetism in the magnetic limit.
The *conservation of total energy* must hold in all inertial reference frames, that is, total energy does not
change with time within a given reference frame. However, the different contributors to total energy, and the
value of the total energy, can vary between inertial reference frames. For example, the kinetic energy of a
particle will depend on its velocity, which varies with inertial reference frame. In contrast, heat energy is
invariant with respect to inertial reference frame because it depends on relative velocities between ions,
electrons and neutrals in motion.
Poynting's theorem is:

$$\frac{\partial W}{\partial t} = -\nabla \cdot \mathbf{S} - \mathbf{J} \cdot \mathbf{E} \tag{14}$$




where $W$ is the energy density of the electromagnetic field:

$$W = \frac{1}{2}\left(\epsilon_0 \mathbf{E} \cdot \mathbf{E} + \frac{1}{\mu_0} \mathbf{B} \cdot \mathbf{B}\right) \tag{15}$$


and $\mathbf{S}$ is the Poynting vector:

$$\mathbf{S} = \mathbf{E} \times \frac{1}{\mu_0} \mathbf{B} \tag{16}$$


Physically, Equation (14) represents energy conservation. It states that the rate of change of
electromagnetic energy density $W$ within a volume equals the energy leaving that volume, via
divergence of Poynting flux ($\nabla \cdot \mathbf{S}$), plus the rate of work done by the electromagnetic field within the
volume ($\mathbf{J} \cdot \mathbf{E}$). In the steady state approximation Poynting's theorem has been used to analyze energy
partitioning at high latitudes (e.g. Equation 5 in Thayer and Semeter, 2004). In this case, $\frac{\partial w}{\partial t} = 0$ and
Poynting's theorem becomes:

$$\nabla \cdot \mathbf{S} = -\mathbf{J} \cdot \mathbf{E} \tag{17}$$


Referring to Table 1, it is clear that none of the individual terms in this equation are GI. Of interest is
whether the equality holds in all inertial reference under the assumptions of Galilean electrodynamics
in the magnetic limit.

It is straightforward to derive how the terms in Equation (17) vary with inertial reference frame using the
transformation equations in Table 1. The Poynting vector $\mathbf{S}$ transforms as:

$$\mathbf{S}' = \mathbf{S} + \frac{1}{\mu_0}(\mathbf{v}_r \times \mathbf{B}) \times \mathbf{B} \tag{18}$$


Using the chain rule applied to vector fields, we find the divergence of $\mathbf{S}$ transforms as:

$$\boldsymbol{\nabla} \cdot \mathbf{S}' = \boldsymbol{\nabla} \cdot \mathbf{S} + \frac{1}{\mu_0}(-(\mathbf{v}_r \times \mathbf{B}) \cdot \boldsymbol{\nabla} \times \mathbf{B} - \mathbf{B} \cdot (\mathbf{v}_r \cdot \boldsymbol{\nabla})\mathbf{B}) \tag{19}$$


where we have used the algebraic rules for the divergence of a cross-product and the curl of a cross
product, the fact that the divergence of $\mathbf{B}$ is zero, and $\mathbf{v}_r$ has zero spatial derivatives. Using Ampere's
law ($\boldsymbol{\nabla} \times \mathbf{B} = \mu_0 \mathbf{J}$) in the steady-state approximation (neglecting displacement currents) we have:



$$\nabla \cdot \mathbf{S}' = \nabla \cdot \mathbf{S} + \mathbf{v}_r \cdot (\mathbf{J} \times \mathbf{B}) - \frac{1}{\mu_0} \mathbf{B} \cdot (\mathbf{v}_r \cdot \nabla)\mathbf{B} \tag{20}$$


The rate at which the electromagnetic field does work ($\mathbf{J} \cdot \mathbf{E}$) transforms as:

$$(\mathbf{J} \cdot \mathbf{E})' = \mathbf{J} \cdot \mathbf{E} - \mathbf{v}_r \cdot (\mathbf{J} \times \mathbf{B}) \tag{21}$$


Combining Equations (17), (20) and (21) we have:

$$\nabla \cdot \mathbf{S}' = -(\mathbf{J} \cdot \mathbf{E})' - \frac{1}{\mu_0} \mathbf{B} \cdot (\mathbf{v}_r \cdot \nabla)\mathbf{B} \tag{22}$$


Equation (22) shows that Poynting's theorem does not hold in the moving inertial reference frame. The
practical consequences of using the widely-used Galilean transformation rules will depend on the specific
situations where the Poynting's theorem is applied.

**2.4 The PGR and Joule Heating**

The PGR can provide physical insight into Joule heating in the context of high latitude electrodynamics. Heat
is a Galilean invariant, since it is due to the random motion of the species independent of their uniform
translational motion. Several authors have expressed the Joule heating term in the energy conservation
equation as (e.g. Cole, 1962; Thayer and Vickrey, 1992; Matsuo and Richmond, 2008):

$$\mathrm{JH} = \mathbf{J} \cdot (\mathbf{E} + \mathbf{V}_n \times \mathbf{B}) = \mathbf{J} \cdot \mathbf{E}^*, \tag{23}$$


where $\mathbf{V}_n$ is the velocity of the neutral wind. (More precisely, $\mathbf{V}_n$ is the mass-weighted velocity of the plasma,
which is dominated by the neutral species).

The discussion in the Appendix of Thayer and Semeter (2004) reaches the same conclusion as in Vasyliūnas
and Song (2005) that heating at high latitudes is due to friction between plasma and neutrals (see also Brekke
and Rino, 1978). Such frictional heating depends on the relative velocity between two species and is thus GI,
consistent with its generating heat, another GI quantity. Thayer and Semeter (2004) note that JH has been
equated to the following quantities: $\mathbf{j} \cdot \mathbf{E}^*$, $\sigma_P E^{*2}$ and $j^2/\sigma_C$ where $\sigma_P$ is the Pedersen conductivity and $\sigma_C$ is
the Cowling conductivity. These expressions are independent of inertial reference frame because they depend
on conductivities that depend on collision frequencies or relative velocities (Leake et al., 2014; Song et al.,
2001). The current density $j$ is assumed to exist in the absence of a net charge density, so $j$ is also reference
frame independent. $\mathbf{E}^*$ refers to the electric field in a particular inertial reference frame and thus is reference



frame independent. However, it should be noted that $\mathbf{E}^*$ is not an electric field and so does not obey Maxwell's
equations.

Strangeway (2012) concludes that Joule dissipation in the reference frame of the neutrals results in heating.
Since heating is GI, this would tend to suggest that Joule dissipation must also be GI. However, Strangeway
(2012) initially defines Joule dissipation as $\mathbf{J} \cdot \mathbf{E}$, which is not GI. We know that $\mathbf{J} \cdot \mathbf{E}$ is not necessarily
dissipative since it is the work done by the electromagnetic field, and can consist of dissipative heating but
also mechanical work done (Matsuo and Richmond, 2008) and the latter is not considered a form of
dissipation.

According to Equation (23), JH can occur in the absence of an electric field (i.e. in an inertial reference frame
where the electric field is zero; see Figure 1). This is consistent with the physical interpretation of Vasyliūnas
(2012) that the "ionospheric current is thus primarily a stress-balance current", and "not an Ohmic current in
the usual sense." The reason the neutral wind velocity appears in Eq. (23) is based on momentum
considerations, as revealed when derived using the full multi-species plasma equations (Brekke and Rino,
1978; Vasyliunas and Song, 2005; Thayer and Vickrey, 1992). When JH occurs in the absence of an electric
field, the currents associated with the heating are generated by ion-neutral velocity differences and not by an
electric field (Mannucci et al., 2018; Vasyliūnas 2012; Thayer and Semeter 2004).
**Discussion**
The PGR is relevant to the physical basis of high latitude electrodynamics. A standard interpretation of
magnetosphere-ionosphere coupling is that magnetospheric currents lead to electric fields that in turn lead to
plasma motion, ion-neutral velocity differences and finally heating (Milan et al., 2017; Cowley, 2000; Kan,
1997). The electric field is not an invariant, so explanations linking currents to electric fields are problematic
from the perspective of the PGR. In addition, Vasyliunas (2001) has shown that electric fields do not cause
bulk plasma motion.

Consistency with the PGR is achieved if ion-neutral velocity differences are viewed as the primary causative
factor of heating (as per Thayer and Semeter (2004) or Vasyliūnas and Song (2005)) and currents (Vasyliūnas
(2012)). In the introduction section of Vasyliūnas and Song (2005) is it stated: "by virtue of the plasma
momentum equation" that the ion-neutral velocity difference is proportional to the current density $\mathbf{J}$. From a
cause and effect perspective, ion-neutral velocity difference *causes* the existence of $\mathbf{J}$. As we show below
from simple momentum considerations, it would not be possible for field-aligned magnetospheric currents
to lead to electric fields that then cause the ions to flow.

The PGR is relevant to explanations for the high velocity ion flows known as Subauroral ion drifts (SAID),
which are often considered as a consequence of Sub-auroral Polarization Stream (SAPS) electric fields. SAPS





are postulated to be the result of magnetospheric currents closing in a low-conductivity region of the
ionosphere, thus leading to large electric fields (Anderson et al., 1993; Clausen et al., 2012). Currents as the
driver for high velocity plasma flows in the lower ionosphere is problematic from the perspective of the PGR,
because the flow velocity depends on inertial reference frame, but the currents do not. The Rice Convection
Model (Wolf et al., 2007; Vasyliunas, 1970) and its variants could be interpreted as implying that currents of
magnetospheric origin closing in the conducting ionosphere are a driver of high latitude convection via
electric fields. From the perspective of the PGR, it is more satisfactory to suggest that large-scale convection
is driven by the velocity differences between the solar wind and magnetospheric plasmas without requiring
an electric field as intermediary in the causal chain.
One might well ask how the velocity of the neutral species (Equation (23)) is relevant when calculating
heating that results from work done by the electromagnetic field, since the neutral species do not experience
the electromagnetic force. A partial answer is that the neutral species carry nearly all of the momentum in
the high latitude ionosphere, and the plasma equations are based in part on considerations of momentum
conservation. The causal chain of forcings relevant to high latitude phenomena would then appear to be:
• Coupling between the solar wind and magnetosphere imparts momentum to the
magnetospheric plasma at altitudes of a few Earth radii.
• Through flux conservation, this momentum is imparted to the magnetospheric plasma at
progressively lower altitudes, eventually down into the ionosphere.
• The momentum imparted to the ionospheric plasma creates a velocity difference between
the plasma and neutral species, leading to transfer of momentum to the neutral species via
collisional processes. The ion-neutral velocity differences lead to the presence of currents
and heating in the ionosphere (Vasyliūnas, 2012; Mannucci et al., 2018).
In this picture, partly alluded to in Vasyliunas and Song (2005), the currents are by-products of momentum
transfer (as the plasma equations suggest) and are not fundamental to the causal chain that couples the
ionosphere to the magnetosphere (see also Vasyliunas, 2012). Currents in the absence of plasma-neutral
collisions do not significantly heat the ionosphere in this picture. The detailed multispecies plasma
calculations in Vasyliunas and Song (2005) show why *the neutral wind velocity is associated with Joule*
*heating in Eq.* (23)*, despite the fact that neutral species are not coupled to electromagnetic forces* (see also
the Appendix of Thayer and Semeter, 2004).
Momentum transfer between ions and neutral species appears central to understanding cause and effect
relationships in magnetosphere-ionosphere coupling (e.g. using the multispecies plasma equations).
Momentum transfer is independent of inertial reference frame, and so can be considered a primary focus for
understanding high latitude processes. Conversely, in the high latitude ionosphere, the electric field depends



on the inertial reference frame, so electric fields as the primary cause of high latitude electrodynamics can be
problematic. In particular, electric fields caused by large-scale field-aligned currents (FACs) closing in the
ionosphere should be reconsidered as a primary driver of high latitude dynamics.

We present an argument, based on momentum conservation, of why electric fields resulting from FACs
closing in the ionosphere are unlikely to *cause* large-scale momentum changes to the ions and neutrals. We
estimate the momentum carried by large-scale high latitude FACs that close horizontally in the ionosphere.
We use an Earth-fixed frame to estimate relative momentum contributions involved in MI-coupling (relative
momentum contributions are independent of reference frame). We focus on the large-scale FACs that carry
most of the current coupling to the magnetosphere, and acknowledge that FACs can occur at multiple scales.
Joule heating that appears simultaneously with large-scale FACs appears to be the dominant factor in energy
transfer from the magnetosphere to the ionosphere (Verkhoglyadova et al., 2017).

FACs originating in the magnetosphere are often carried by electrons due to their high mobility (Carlson et
al., 1998; Sugino et al., 2002). For typical large-scale disturbance FACs of approximately $\sim 2 \ \mu A/m^2$ (e.g.
Iijima and Potemra, 1976), and assuming an Earth-fixed frame, electron velocities are in the range ~156 m/s,
assuming typical densities of $8 \times 10^{10}$ el/m$^3$ in the lower ionosphere (near 110 km). These velocities are
low enough that relativistic effects are not needed, i.e. the PGR is valid. Larger charged particle densities,
e.g. created by intense electron precipitation, imply lower electron velocities for the same current.

Since the electron current $\mathbf{j}_e$ is given by $\mathbf{j}_e = qN_e\mathbf{u}_e$ and electron momentum density $\mathbf{p}_e$ per unit volume is
given by $\mathbf{p}_e = m_eN_e\mathbf{u}_e$, we have that $\mathbf{p}_e = m_e\mathbf{j}_e/q$, where $m_e$ is the electron mass. Thus, momentum density
of the FACs is approximately given by: $\mathbf{p}_e \sim 1.1 \mathrm{x} 10^{-17}$ kg-m-s$^{-1}$-m$^{-3}$. We compare this momentum density
residing with electron currents to changes in momentum density of the neutral species $\mathbf{p}_n$ that occur during
disturbed periods. We assume that: 1) neutral densities are 100 times or more larger than ion densities as is
typical in the lower ionosphere, 2) the neutral species are dominated by atomic oxygen and 3) velocity
changes of the neutrals under disturbed conditions can be comparable to those of the electrons (~156 m s$^{-1}$)
(e.g. Zhang and Shepherd, 2002). These assumptions apply to the lower ionosphere where collisions between
ions and neutrals are important. Under these assumptions, we find that the change in neutral momemtum due
to disturbed conditions is $\Delta\mathbf{p}_n \sim 3.3 \mathrm{x} 10^{-11}$ kg-m-s$^{-1}$-m$^{-3}$, which is several orders of magnitude larger than the
momentum density carried by electronic field-aligned currents.

It appears that during disturbed conditions, the momentum change of the neutral species dominates by orders
of magnitude the momentum carried by the disturbance-related currents. This would remain true even when
the currents are carried by ions, since from the magnetosphere these currents would be carried by light ions
(hydrogen) versus the dominant heavier neutrals (e.g. oxygen) at altitudes where collisions occur. Velocity
changes for the neutral species can easily reach 100 m/s or larger at high latitudes during disturbed conditions





(Kosch et al., 2010). We conclude that momentum changes associated with neutrals dominate over
momentum carried by FACs entering from the magnetosphere that "close" as horizontal currents in the
ionosphere. The changes in neutral momentum are thus due to collisions with ions, not due to the field aligned
currents.
**Conclusion**
We have reviewed electromagnetism in the context of high latitude electrodynamics to show that commonly-
used relationships that transform electric and magnetic fields between inertial frames correspond to a limiting
case known as the "magnetic limit" of Galilean electromagnetism. This limit is used in the literature related
to magnetosphere-ionosphere coupling (Equations (3) and (4)). The magnetic limit of special relativity
applies when electric and magnetic field magnitudes are related by $cB \gg E$ which is valid in the collisional
region of the high latitude ionosphere (altitudes near 110 km).

The Principle of Galilean Relativity is used to gain insight into the physical basis for magnetosphere-
ionosphere coupling. We have considered Ohm's law from the perspective of the PGR and noted that Ohm's
law need not be a relationship between currents and electric fields. We suggest that physical insight is gained
by considering an inertial reference frame that is not tied to a particular species of the material medium: the
reference frame in which the large-scale electric field is zero. We have also considered the steady-state
formulation of Poynting's Theorem from the perspective of the PGR and shown that, in the magnetic limit
of Galilean electromagnetism, Poynting's Theorem does not necessarily hold in all inertial reference frames.
Previous authors have noted that Galilean electromagnetism is not consistent with the full set of Maxwell's
equations: the displacement current term must be removed to achieve consistency.

We have used the PGR to gain insight into why Joule heating at high latitudes is due to friction between
plasma and neutrals, as shown by detailed multi-species plasma calculations (Brekke and Rino, 1978,
Vasyliunas and Song, 2005 and Thayer and Semeter, 2004). Despite the term "Joule heating" that is
associated with experiments where currents generated by electric fields cause heating, high latitude heating
cannot depend on electric fields, the latter being reference frame-dependent. Plasma motion relative to the
neutrals is of course a consequence of electromagnetic forces, originating in relative motion between
planetary plasma and solar wind plasma in the collisionless regime of the interplanetary medium.

The Galilean transformation rules for currents and electric fields are very different from each other. This
suggests it is problematic to assert that currents of magnetospheric origin *cause* horizontal electric fields in
the ionosphere. Such a causal relation would seem to be reference frame dependent. At E-layer altitudes in
the ionosphere, currents are independent of inertial reference frame, whereas perpendicular electric fields are
not. Thus it is problematic to assert that the cause of high-velocity plasma flows known as SAID are
horizontal currents closing in the ionosphere. During disturbed conditions generally, the momentum carried

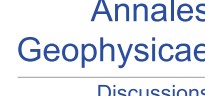

by large-scale currents of magnetospheric origin is much less than the momentum changes of the neutral
species occurring during disturbed conditions.

**Acknowledgements**
This research was carried out at the Jet Propulsion Laboratory, California Institute of Technology, under a
contract with the National Aeronautics and Space Administration. Government sponsorship acknowledged.
One of the authors (AJM) wishes to acknowledge constructive discussions with Professor Paul Song,
University of Massachusetts at Lowell. We acknowledge sponsorship of the NASA and National Science
Foundation Partnership for Collaborative Space Weather Modeling and to the NASA Living With a Star
Science program. No data were used in this paper.

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





**Figures**

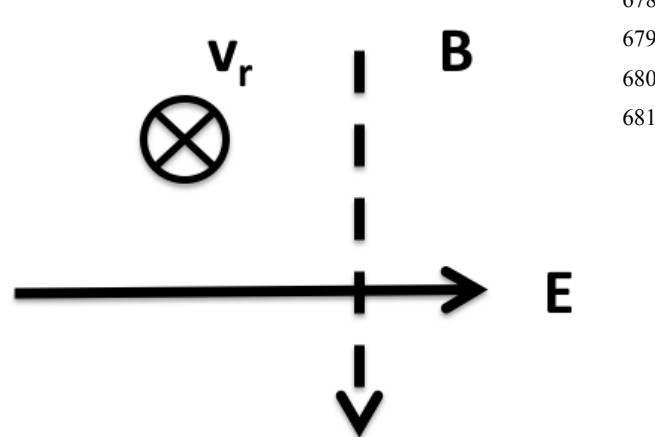





**Figure 1. Schematic representation of electric and magnetic fields at northern high latitudes. Magnetic fields are directed vertically (B). Electric fields are directed horizontally (e.g. in an Earth fixed frame, resulting from magnetospheric convection). In a reference frame moving with velocity $v_r$ in the direction shown (into the page), the electric field is zero if the reference frame moves with speed $\|E\|/\|B\|$. For "typical" disturbed conditions, non-relativistic speeds of ~ 1.6 km/s are sufficient.**