# Peer review of "Using the Galilean Relativity Principle to Understand the"

_Annales Geophysicae, 2019_

## Referee Comment (RC1) · Anonymous Referee #1 · 11 Oct 2019

Report on the manuscript angeo-2019-108
"Using the Galilean Relativity Principle to Understand the Physical Basis for Magnetosphere-Ionosphere Coupling Processes"
by Mannucci et al.

This reader agreed to review this manuscript because the title sounded very interesting, but the manuscript really had limited new information in it. This review is not of the opinion that this manuscript should not be published (it does in fact make the reader think), but the manuscript could be improved with more focus on attaining information relevant to its title, and more focus on being accurate in its discussion of the behavior of high-latitude ionospheric electrodynamics.

This manuscript fails to fulfill the promise of its title: the manuscript says what is not correct (which has been said before), but does not say what physical processes form the basis for magnetosphere-ionosphere couple. I.e. it is never explained how the magnetosphere drives ionospheric motion and currents (which co-exist). The closest it comes to an explanation is saying that momentum is transferred from the magnetosphere to the ionosphere (Lines 429-436).

What has been said before is (a) that the perpendicular current is caused by the relative motion of plasma and neutrals across the magnetic field, (b) that Joule heating is associated with the work done to move charged particles through the neutrals (cf. the Drude model), and (c) explanations for ionospheric electrodynamics exist that do not focus on electric fields.

This reviewer is aware that clear pictures of cause and effect in plasma electrodynamics are difficult because Maxwell's equations describe consistency, not cause and effect. One should look at the mechanisms that extract energy (and momentum), that convert energy and momentum, and that transport energy and momentum. It seems unwise to use PGR, which only sometimes works, to organize a discussion describing the driving of currents, the transfer of momentum, and the transfer of energy. As Section 2.3 notes, Poynting's theorem does not hold in PGR. Perhaps this is why the title of the article is not fulfilled. The concepts of PGR seem more like handcuffs than tools.

Several times in the manuscript it is stated that "something" is inconsistent with PGR (e.g. lines 164, 166, 304): it is hard to tell if this is a criticism of the "something" or a criticism of PGR.

Lines 165-167: The statement "electric fields that arise due to charge accumulation are inconsistent with the usual field transformation equations…" is not true. For example, there is charge density in the ionosphere wherever there is plasma vorticity that is parallel to the magnetic field (i.e. a shear in a perpendicular flow $v_{perp}$), and that charge density cannot be transformed away. There is an electric field (and a nonzero divergence of the electric field) associated with that charge density (and associated with the shear flow). The motion of the charge density does make a perturbation to B, but its magnitude is $\Delta B = (v_{perp}/c)^2 B$: it is consistent with PGR to ignore this change in B.

Lines 199-207: This reader is uncertain as to the purpose of this paragraph. Landau damping involves an electric field that is parallel to the velocity of interest and that parallel electric field does not change with reference frame as you go to the particle's frame.

Line 225: "For electromagnetic waves," should be "For electromagnetic waves in vacuum,"

Lines 238-241: This reader is completely confused by this paragraph.

Lines 304-306: The statement "requires the creation of polarization charge, which is inconsistent with the Galilean invariance of the magnetic field" is not correct. Polarization charge is in fact created at the edges any spatially limited flow (plug in Coulomb's law for a shear flow where the flow is described by ExB drift). As noted above, advection of such charge density creates a perturbation to be that is of order $(v/c)^2B$ (where v is the advection velocity), which is ignorable in PGR and not "inconsistent" with PGR.

Lines 459-480: This is a misleading calculation: no one would believe that the parallel-to-B momentum of the magnetospheric charge carriers has anything to do with the perpendicular-to-B momentum of the ionosphere. This is like calculating the momentum of electrons in a wire to explain momentum provided by an electric fan or the momentum of an electric car. Further, what does this calculation have to do with an electric field? The way to look at the momentum change of the ionosphere is to look at the surface integral of the Maxwell stress tensor, but since Poynting's theorem is not valid in PGR, this might not work in present study.

Lines 521-522: "Thus it is problematic to assert that the cause of high velocity plasma flows known as SAID are horizontal currents closing in the ionosphere".  You are saying what it is not the cause, but never saying what is the cause!

The manuscript repeatedly criticizes the electric field because it differs from frame to frame, asserting therefore that it cannot be the explanation (cause) of things. One should assume that if a problem is worked correctly in two different reference frames, one should get two answers that agree. (Principle of invariance of proper physical description in relativity.) Maybe PGR is not the right way to think about high-latitude electrodynamics.

Here is a gedanknenexperiment in support of the ionospheric electric field. Call it the cold-Earth problem. Let's say that the high-latitude ionospheric plasma and neutrals are absolutely cold (no collisions) and that the ions, electrons, and neutrals are all at rest (no wind). The one frame that makes sense to use to look at this is the frame of the observer on the ground, the neutrals, the ions, and the electrons (all the same frame). If from this cold start I want to get the plasma to flow relative to the neutrals, and to get a horizontal current that is associated with the plasma flow through the neutrals, by charged-particle orbit theory I must get a horizontal electric field into the ionosphere to start the ExB drifting of the ions and electrons, which collisions with the neutrals will disrupt and form a Pedersen and/or Hall current. When everything is at rest, there can be no other force on a charged particle than the electric field force. The explanation of how the magnetosphere drives the ionosphere in this cold-Earth gedankenexperiment is explaining how the magnetosphere got that needed electric field into the ionosphere.

This reviewer looks forward to reading an improved draft. (Thinking is good!)

---

## Author Comment (AC1) · 22 Nov 2019

Report on the manuscript angeo-2019-108
"Using the Galilean Relativity Principle to Understand the Physical Basis for
Magnetosphere- Ionosphere Coupling Processes"
by Mannucci et al.

This reader agreed to review this manuscript because the title sounded very interesting, but the manuscript really had limited new information in it. This review is not of the opinion that this manuscript should not be published (it does in fact make the reader think), but the manuscript could be improved with more focus on attaining information relevant to its title, and more focus on being accurate in its discussion of the behavior of high-latitude ionospheric electrodynamics.

This manuscript fails to fulfill the promise of its title: the manuscript says what is not correct (which has been said before), but does not say what physical processes form the basis for magnetosphere-ionosphere coupling. I.e. it is never explained how the magnetosphere drives ionospheric motion and currents (which co-exist).

We appreciate these comments from the reviewer. In the following we attempt to clarify our explanations.

The closest it comes to an explanation is saying that momentum is transferred from the magnetosphere to the ionosphere (Lines 429-436).

We here elaborate on this explanation. The process by which momentum is transferred from the magnetosphere to the ionosphere is the same process by which momentum is carried in a particular direction by the solar wind. The solar wind consists of a non-collisional, magnetized, fully ionized plasma. The momentum is largely carried by the ions due to their larger mass compared to the electrons. Due to its non-collisional nature, the only interactions between the ions and electrons are electrodynamic. These electrodynamic forces lead to a collective motion of the plasma away from the sun towards the Earth. Without such forces, the random motion of the plasma would disperse the plasma and it would not travel collectively, carrying momentum in a well-defined direction. When this directed plasma encounters the magnetosphere, momentum is imparted via electrodynamic forces to magnetospheric ions, and eventually down towards the ionosphere until at lower altitudes collisional forces begin to play a role.

What has been said before is (a) that the perpendicular current is caused by the relative motion of plasma and neutrals across the magnetic field, (b) that Joule heating is associated with the work done to move charged particles through the neutrals (cf. the Drude model), and (c) explanations for ionospheric electrodynamics exist that do not focus on electric fields.

We agree that (a) has been stated previously. We would appreciate references to the Drude model so that we can reference it in the context of this paper. We would appreciate references for (c). Although explanations may exist that do not rely on electric fields, there are explanations that do rely on electric fields. We are addressing those explanations that do rely on electric fields.

This reviewer is aware that clear pictures of cause and effect in plasma electrodynamics are

difficult because Maxwell's equations describe consistency, not cause and effect.

We appreciate this comment. We believe cause and effect can be inferred to some extent from Maxwell's equations (ME) because ME determine how electric and magnetic fields originate from their sources (charges and currents). Electric and magnetic fields determine the forces on charges and currents, thus leading to various effects.

One should look at the mechanisms that extract energy (and momentum), that convert energy and momentum, and that transport energy and momentum. It seems unwise to use PGR, which only sometimes works, to organize a discussion describing the driving of currents, the transfer of momentum, and the transfer of energy. As Section 2.3 notes, Poynting's theorem does not hold in PGR. Perhaps this is why the title of the article is not fulfilled. The concepts of PGR seem more like handcuffs than tools.

We appreciate this comment, but do not completely agree with all of it. The PGR "always works" in the sense that the principle of special relativity (PSR) is absolute: the laws of physics are invariant to reference frame. The approximate form of special relativity as embodied in the PGR is not always consistent with ME. In this paper, we use those aspects of the PGR that are useful, despite its imperfection. We believe the paper may be clearer if we refer to the PSR, and not always to the PGR. We can do so in a revision. We propose a revised title to address this comment: "A discussion of physical processes in magnetosphere-ionosphere coupling, including the problematic use of Galilean relativity."

Transfer of energy and momentum is relevant to the PSR because the PSR suggests that the electric field is not always a suitable explanation for such transfer. This is described in the paragraph lines 411-421.

We agree that the discussion of momentum transfer (lines 429-436) is not well motivated in the current version. We address this below (under your comment for lines 459-480).

Several times in the manuscript it is stated that "something" is inconsistent with PGR (e.g. lines 164, 166, 304): it is hard to tell if this is a criticism of the "something" or a criticism of PGR.

This problem is addressed by de-emphasizing PGR in favor of the PSR. All the physics must be consistent with the PSR. This could be achieved in the revision.

Lines 165-167: The statement "electric fields that arise due to charge accumulation are inconsistent with the usual field transformation equations…" is not true. For example, there is charge density in the ionosphere wherever there is plasma vorticity that is parallel to the magnetic field (i.e. a shear in a perpendicular flow $v_{perp}$), and that charge density cannot be transformed away. There is an electric field (and a nonzero divergence of the electric field) associated with that charge density (and associated with the shear flow). The motion of the charge density does make a perturbation to B, but its magnitude is $\otimes B = (v_{perp}/c)^2 B$: it is consistent with PGR to ignore this change in B.

The presence of non-zero charge density is inconsistent with the widely-used transformation rule that current density is invariant to reference frame (see the text above Equation (4) in

Thayer and Semeter, 2004). To the best of our understanding, the perturbation to the magnetic field is not second-order. If a charge density $\rho$ exists in the reference frame at rest, then in the moving reference frame the current density originating from this charge density is $-\rho\mathbf{v_r}$ (see Equation (7) of this paper). This current density in the moving frame will lead to a magnetic field ($\nabla \times \mathbf{B} = \mu_0\mathbf{J}$), in contradiction to our Equation (4) which is part of the PGR.

Lines 199-207: This reader is uncertain as to the purpose of this paragraph. Landau damping involves an electric field that is parallel to the velocity of interest and that parallel electric field does not change with reference frame as you go to the particle's frame.

We agree this paragraph is somewhat out of context. A previous reviewer had wanted us to include these specific examples of the use of PGR in geospace. We can remove this paragraph in the revision.

Line 225: "For electromagnetic waves," should be "For electromagnetic waves in vacuum,"

Our intention in this sentence was to refer to the hypothetical aether as the "preferred reference frame", that was postulated as the material medium in which electromagnetic waves propagated. The Michelson-Morley experiment is evidence that this aether does not exist. In the context of sound waves, an analogous experiment would show the presence of a material medium for sound waves. We could simply remove this sentence.

Lines 238-241: This reader is completely confused by this paragraph.

We regret that this paragraph is not more clear. It is perhaps sufficient to simply append the following sentence to the end of the previous paragraph and otherwise delete this paragraph.
* * *
The second term in the expression for $\mathbf{E}^*$ has been referred to as a dynamo electric field when $\mathbf{V} = \mathbf{V_n}$, the velocity of the neutral species (Richmond, 1995), although this term does not refer to a physical electric field.
* * *
Lines 304-306: The statement "requires the creation of polarization charge, which is inconsistent with the Galilean invariance of the magnetic field" is not correct. Polarization charge is in fact created at the edges any spatially limited flow (plug in Coulomb's law for a shear flow where the flow is described by ExB drift). As noted above, advection of such charge density creates a perturbation to be that is of order $(v/c)^2B$ (where v is the advection velocity), which is ignorable in PGR and not "inconsistent" with PGR.

As we have stated above, the presence of a charge density $\rho$ in one reference frame creates a current density in the moving reference frame ($\rho\mathbf{v_r}$). The current density in the moving frame should create a magnetic field according to Maxwell's equation. This is not a second-order effect in our view. See Equation (7) of the paper. We can be more explicit in the revision.

Lines 459-480: This is a misleading calculation: no one would believe that the parallel-to-B momentum of the magnetospheric charge carriers has anything to do with the perpendicular-to-B momentum of the ionosphere. This is like calculating the momentum of electrons in a wire to explain momentum provided by an electric fan or the momentum of an electric car. Further, what does this calculation have to do with an electric field? The way to look at the momentum change of the ionosphere is to look at the surface integral of the Maxwell stress tensor, but since Poynting's theorem is not valid in PGR, this might not work in present study.

We agree with your statement that the momentum of electrons in a wire do not explain the momentum of a car. The car experiences a force between the road and tires that changes the car's momentum. If the car were an isolated system, a momentum analysis concerning the electrons in the wire might be relevant. In the current paper version, we do not clearly present an analysis of what external forces are acting on different parts of the system (electrons, ions, neutrals), so the momentum calculation can appear misleading and we will remove it.

We appreciate the comment: "what does this calculation have to do with an electric field?" This leads us to consider how field-aligned currents are related to electric fields in the ionosphere. This relationship is directly relevant to the PGR, because currents and fields have different transformation properties. The relevant literature is that which discusses models that connect the cross-polar cap potential to region 1 currents. The cross-polar cap potential is due to the large-scale convection electric field.

A model that relates region 1 currents directly to the convection electric field is discussed in Siscoe et al. (2002; doi: 10.1029/2001JA000109) where the introduction states "$\Phi_m$ (magnetospheric convection potential) is then impressed via equipotential magnetic field lines onto the ionosphere, where it becomes the $\Phi_{pc}$ (transpolar ionospheric potential) that generates region 1 currents." This statement suggests a causative relation between electric field and region 1 currents and thus is relevant to our discussion of the PGR. In a revision, we will reference Siscoe et al. (2002) and papers that use this model (Rothwell and Jasperse, 2006 and 2007) and discuss these models in the context of the PGR.

This type of model does not account for the neutrals, which is a key missing element. As we and others argue (e.g. Mannucci et al., 2018; Cowley, 2000) the ionospheric currents are the result of plasma flowing relative to the neutrals. Divergence of these currents can lead to field-aligned currents into the magnetosphere. Processes in the magnetosphere can lead to field-aligned currents also, that enter the ionosphere. Overall, we agree that the momentum discussion does not bring clarity to understanding these processes.

Lines 521-522: "Thus it is problematic to assert that the cause of high velocity plasma flows known as SAID are horizontal currents closing in the ionosphere". You are saying what it is not the cause, but never saying what is the cause!

We are not prepared to state the cause. However, we believe it is useful, using relativistic arguments, to rule out causes that have been proposed.

The manuscript repeatedly criticizes the electric field because it differs from frame to frame,

asserting therefore that it cannot be the explanation (cause) of things. One should assume that if a problem is worked correctly in two different reference frames, one should get two answers that agree. (Principle of invariance of proper physical description in relativity.) Maybe PGR is not the right way to think about high-latitude electrodynamics.

We should not be leaving the impression that we are criticizing the electric field. We are merely noting that its transformation properties are important to consider in light of understanding physical processes. We believe that PGR is useful in that it can guide how explanations will differ depending on reference frame. We will add text to clarify this point.

Here is a gedanknenexperiment in support of the ionospheric electric field. Call it the cold-Earth problem. Let's say that the high-latitude ionospheric plasma and neutrals are absolutely cold (no collisions) and that the ions, electrons, and neutrals are all at rest (no wind). The one frame that makes sense to use to look at this is the frame of the observer on the ground, the neutrals, the ions, and the electrons (all the same frame). If from this cold start I want to get the plasma to flow relative to the neutrals, and to get a horizontal current that is associated with the plasma flow through the neutrals, by charged-particle orbit theory I must get a horizontal electric field into the ionosphere to start the ExB drifting of the ions and electrons, which collisions with the neutrals will disrupt and form a Pedersen and/or Hall current. When everything is at rest, there can be no other force on a charged particle than the electric field force. The explanation of how the magnetosphere drives the ionosphere in this cold-Earth gedankenexperiment is explaining how the magnetosphere got that needed electric field into the ionosphere.

This is an interesting experiment. We would describe the processes as follows. To initiate plasma flow relative to neutrals requires imparting momentum to the ions. As we suggest above (response lines 429-436), the momentum transfer occurs via coupling to a moving plasma – the solar wind plasma. The solar wind plasma couples to the magnetospheric plasma, and eventually to the ionospheric plasma, via collisionless processes – the same processes that are responsible for transferring momentum in the solar wind to the outer magnetosphere. Conversely, as Vasyliunas has shown (2001, doi:10.1029/2001GL013014), electric fields imposed on a plasma do not cause that plasma to flow.

This reviewer looks forward to reading an improved draft. (Thinking is good!)

The editor has suggested that the discussion can proceed without the revised draft. This makes sense because we are not yet in full alignment regarding how the paper should proceed forward. We certainly look forward to providing a revised draft once we know how to proceed.

---

## Referee Comment (RC2) · Anonymous Referee #2 · 23 Nov 2019

Review report on "Using the Galilean Relativity Principle to understand the physical basis for magnetosphere-ionosphere coupling processes" by Anthony J. Mannucci et al.

General comments

The authors discuss the problematic interpretation of the electric field in ionospheric electrodynamics, using the "Galilean Relativity Principle" which is a limiting case of the theory of special relativity. While this assertion seems reasonable, the paper includes unacceptable reasoning as I comment below. Also the content of the paper is generally "reinterpretation" of the previous work, without much of scientifically new. However, I think the problem presentation by the authors is useful to the space science community, because such discussion cannot be found in textbooks nor in journal papers. I would like to recommend publication after revision in response to the following comments.

Specific comments

(1) Although I agree with the authors on the conventional problematic interpretation of ionospheric electrodynamics in terms of the electric field, I disagree with the authors' conclusion that the magnetospheric currents that close in the ionosphere play no role in the magnetosphere-ionosphere coupling (on lines 25-27, lines 414-415, lines 438-440, lines 454-490, lines 522-524). This conclusion is drawn on lines 452-490 by comparing the kinetic momentum of current-carrying electrons with the kinetic momentum of neutrals. I cannot understand the authors' logic employed. The density of electromagnetic momentum carried by the field-aligned currents is given by

$$\mathbf{p}_m = \frac{\mp \dfrac{1}{\mu_0} B_0 \Delta \mathbf{B}_\perp}{V_A} = \mp \sqrt{\frac{m_i n_i}{\mu_0}} \Delta \mathbf{B}_\perp \qquad \left( \begin{array}{l} \text{the upper minus sign applies to the northern ionosphere} \\ \text{the lower plus sign applies to the southern ionosphere} \end{array} \right)$$

and can be comparable to the momentum density of neutrals ( $\Delta \mathbf{p}_n$ ). In fact, in the case of a magnetospheric dynamo, what causes a nonzero relative velocity between ions and neutrals ( $\mathbf{u}_i - \mathbf{u}_n$, which is initially zero) is the ion acceleration by the $\mathbf{j} \times \mathbf{B}$ force in the ionosphere. The ions are accelerated until the $\mathbf{j} \times \mathbf{B}$ force balances the collisional force $-m_i n_i \nu_{in} \left( \mathbf{u}_i - \mathbf{u}_n \right)$.

(2) In all descriptions in this paper, an initial finite $\mathbf{u}_i - \mathbf{u}_n$ with $\mathbf{j} = \mathbf{0}$ seems assumed. This is the case of ionospheric dynamo (neutral dynamo). At the same time, however, the case of magnetospheric dynamo (magnetospheric flow increases suddenly while $\mathbf{u}_i - \mathbf{u}_n = \mathbf{0}$ and $\mathbf{j} = \mathbf{0}$ in the ionosphere) is often discussed in parallel or in mixture. This is very confusing, and sometimes the reasoning is incorrect. Examples are on lines 291-292, lines 419-421, lines 429-436, and lines 451-452. The authors should separate the discussion.

(3) (Minor comment) On line 431, "Through flux conservation": What kind of flux do the authors mean?

(4) (Minor comment) I cannot understand the statement on lines 506-507. In what context did the authors add this statement?

Technical comments

None. The manuscript is well written.

---

## Author Comment (AC2) · 7 Dec 2019

Review report on "Using the Galilean Relativity Principle to understand the physical basis for magnetosphere-ionosphere coupling processes" by Anthony J. Mannucci et al.

General comments

The authors discuss the problematic interpretation of the electric field in ionospheric electrodynamics, using the "Galilean Relativity Principle" which is a limiting case of the theory of special relativity. While this assertion seems reasonable, the paper includes unacceptable reasoning as I comment below. Also the content of the paper is generally "reinterpretation" of the previous work, without much of scientifically new. However, I think the problem presentation by the authors is useful to the space science community, because such discussion cannot be found in textbooks nor in journal papers. I would like to recommend publication after revision in response to the following comments.

Specific comments

(1) Although I agree with the authors on the conventional problematic interpretation of ionospheric electrodynamics in terms of the electric field, I disagree with the authors' conclusion that the magnetospheric currents that close in the ionosphere play no role in the magnetosphere-ionosphere coupling (on lines 25-27, lines 414-415, lines 438-440, lines 454-490, lines 522-524).

We agree that we should not be suggesting that these currents play no role in the coupling. Rather they are important "by-products of momentum transfer" that occurs for other reasons than the currents themselves. We discuss this topic further in response to the referee's next comment, where we seek clarification. In the revision, we should remove language that suggests "no role". Lines 414-415 could be removed. In lines 438-440, we should simply remove the phrase "and are not fundamental to the causal chain that couples the ionosphere to the magnetosphere". For lines 545-490, we are referring specifically to the momentum argument, which is further discussed below. We should modify the last sentence in this part to read as follows: "We conclude that momentum changes associated with neutrals are larger than the momentum carried by FACs entering from the magnetosphere…The changes in neutral momentum are due to collisions with ions, not due to the field aligned currents." The lines 522-524 can remain if our momentum arguments are correct and relevant.

This conclusion is drawn on lines 452-490 by comparing the kinetic momentum of current-carrying electrons with the kinetic momentum of neutrals. I cannot understand the authors' logic employed. The density of electromagnetic momentum carried by the field-aligned currents is given by

$$\mathbf{p}_m = \frac{\mp \dfrac{1}{\mu_0} B_0 \Delta \mathbf{B}_\perp}{V_A} = \mp \sqrt{\frac{m_i n_i}{\mu_0}} \Delta \mathbf{B}_\perp \quad \left( \begin{array}{l} \text{the upper minus sign applies to the northern ionosphere} \\ \text{the lower plus sign applies to the southern ionosphere} \end{array} \right)$$

and can be comparable to the momentum density of neutrals ($\Delta \mathbf{p}_n$). In fact, in the case of a magnetospheric dynamo, what causes a nonzero relative velocity between ions and neutrals ($\mathbf{u}_i - \mathbf{u}_n$, which is initially zero) is the ion acceleration by the $\mathbf{j} \times \mathbf{B}$ force in the ionosphere. The ions are accelerated until the $\mathbf{j} \times \mathbf{B}$ force balances the collisional force $-m_i n_i \nu_{in} (\mathbf{u}_i - \mathbf{u}_n)$.

We appreciate this comment from the referee. We would benefit from a reference to the above equation for $\mathbf{p}_m$ so that we are sure to understand its derivation. Electromagnetic momentum is usually proportional to $\sim \mathbf{E} \times \mathbf{B}$ (e.g. Jackson, 1975, equation 6.125). The above expression might be derived by relating the electric field to a current that generates the perturbation magnetic field. However, we are not sure where the perturbation magnetic field is evaluated in this expression. The perturbation magnetic field will in general not be spatially uniform?

We note that FACs might also carry momentum due to the motion of the particles themselves, and not solely via the electromagnetic field. Momentum conservation is discussed in the paper by Vasyliunas (Annales Geophysicae, 2007 doi: 10.5194/angeo-25-255-2007) starting with momentum conservation (their Equation (1)). The linear momentum per unit volume is defined (their Equation (2)) as consisting of an electromagnetic term and a term proportional to the mass density and bulk flow of the medium. Just above their Equation (3), it is stated that "Under the usual assumptions of charge quasi-neutrality of the plasma, nonrelativistic bulk flows, and Alfven speed [much less than the speed of light], the electromagnetic contribution to the linear momentum density (second term on the right-hand side of Eq. (2)) and the electric-field terms in the Maxwell stress tensor can be neglected." We are not suggesting that electromagnetic momentum can be ignored in the specific example of FACs that the referee has mentioned. However, we are seeking guidance on this subject.

Our own calculation of the term $\mathbf{p}_m$ above yields a value of $\sim 4 \times 10^{-12}$ using a perturbation magnetic field value of 100 nT and other values listed in the paper (lines 463-480). This is about a factor of 10 smaller than the representative value we use in the paper for the change in neutral momentum ($\sim 3 \times 10^{-11}$). However, we agree that larger perturbation magnetic fields have been observed ($\sim 1000$ nT) that could result in comparable values between $\mathbf{p}_m$ and the neutral momentum change. Thus we are very interested to have more details on the derivation of $\mathbf{p}_m$.

We do not understand the comment regarding the magnetospheric dynamo. The jxB force is the result of the magnetic component of the Lorentz force. The magnetic component of the Lorentz force does not increase the speed of plasma particles. It only changes their direction. So, we are not sure how the jxB force results in accelerating the ions so that they achieve speeds to balance the collisional force.

(2) In all descriptions in this paper, an initial finite $\mathbf{u}_i - \mathbf{u}_n$ with $\mathbf{j} = 0$ seems assumed. This is the case of ionospheric dynamo (neutral dynamo). At the same time, however, the case of magnetospheric dynamo (magnetospheric flow increases suddenly while $\mathbf{u}_i - \mathbf{u}_n = 0$ and $\mathbf{j} = 0$ in the ionosphere) is often discussed in parallel or in mixture. This is very confusing, and sometimes the reasoning is incorrect. Examples are on lines 291-292, lines 419-421, lines 429-436, and lines 451-452. The authors should separate the discussion.

Thanks to the referee for pointing out this potential source of confusion. We should clarify the presentation. In lines 291-292, we are referring to the "ionospheric dynamo" without reference to how the initial velocity difference was created. Ultimately, it must have arisen due to increased magnetospheric flow. We could remove this statement, as it is not really needed. We also agree that lines 451-452 are may cause confusion and can be removed.

(1) (Minor comment) On line 431, "Through flux conservation": What kind of flux do the authors mean?

Thanks to the referee for pointing this out. In a revision, we will refer to the "frozen-in flux" condition that occurs in collisionless plasmas (as described in Bellan's text on plasma physics). The frozen-in flux condition is why the plasma can move collectively and transfer momentum from one location to another, despite there being no collisions between the constituent particles.

(2) (Minor comment) I cannot understand the statement on lines 506-507. In what context did the authors add this statement?

We appreciate that the referee pointed this out. This statement should be explained in more detail. It is based on the literature on Galilean electromagnetism. The context can be found, for example, in Preti et al. (2009). Preti et al. apply the "standard" Galilean transformation equations (our Equations (3) and (4)) to the four Maxwell's equations in differential form (Preti's Section 3.2). The conclusion is that Gauss' and Ampere's laws (divergence of E, and curl of B, respectively) are not invariant under the Galilean transformation. This contradicts relativistic invariance, demonstrating the problematic nature of Galilean electromagnetism as typically applied. We can expand on this point in the revision.

Technical comments

None. The manuscript is well written.